# Methodological Considerations for a Risk Model Adopted into the Chronic Disease Prevention Policy of Taiwan. Comment on Chang et al. Developing and Validating Risk Scores for Predicting Major Cardiovascular Events Using Population Surveys Linked with Electronic Health Insurance Records. *Int. J. Environ. Res. Public Health* 2022, *19*, 1319

**DOI:** 10.3390/ijerph22071113

**Published:** 2025-07-15

**Authors:** Che-Jui Chang

**Affiliations:** 1Institute of Occupational and Environmental Health Sciences, National Taiwan University College of Public Health, Taipei 100, Taiwan; jcred@ntuh.gov.tw; 2Department of Family Medicine, National Taiwan University Hospital Hsin-Chu Branch, Hsinchu 302, Taiwan

Chang, H.-Y. et al. (2022) developed a risk prediction model for major adverse cardiovascular events (MACEs), coronary heart disease (CHD), and stroke using nationwide claims data retrieved from the Taiwan National Health Insurance (NHI) records [1]. This model has been integrated into the “National Chronic Disease Prevention Policy” of Taiwan as part of a risk calculation platform managed by the Health Promotion Administration (HPA) (https://cdrc.hpa.gov.tw/, accessed on 14 May 2025). It is now mandated for use in adult health check-ups across medical institutions to estimate the 10-year risk for five chronic diseases: MACEs, CHD, stroke, diabetes, and hypertension [2]. Given its broad application in clinical care, public health interventions, and resource planning, ensuring clarity of the methodology is essential for scientific transparency, reproducibility, and public trust, in line with the TRIPOD guidelines for reporting prediction models [3].

Although the time-to-event nature of the data suggests that a Cox proportional hazards model may have been used, the paper does not explicitly confirm the regression model applied. While Harrell’s C-index and the Akaike information criterion (AIC) are mentioned—metrics typically associated with survival analysis—the absence of a clear description of the model limits independent validation and interpretability.

Table 2 in the Chang report presents covariate coefficients for MACE, CHD, and stroke. However, calculating absolute 10-year risk using a Cox model requires the baseline survival probability S_0_(10). The paper reports the “S_0_(10)” for MACE as 0.82 for men and 0.90 for women, but the description is inconsistent. The authors initially refer to these as “10-year event probabilities,” which typically denote 1–S(10), yet subsequently add further that “82% of men and 90% of women remained MACE-free,” implying S(10). While the latter is more consistent with survival probability, it remains unclear whether this estimate reflects the Cox model’s baseline survival (i.e., with all covariates set to zero) or a population-level figure. The lack of clarity on this point, combined with the absence of S_0_(10) values for CHD and stroke, challenges replicating the model and applying it in risk estimation.

In addition, although the model is used by the HPA platform to assess the risk of diabetes and hypertension, the Chang paper does not describe the development or validation of prediction models for these outcomes. This raises concerns about generalizability, as clinicians are expected to apply the tool to five conditions, but the peer-reviewed documentation covers only three (i.e., MACE, CHE, and stroke).

Moreover, the paper does not discuss key statistical considerations, such as model assumptions, diagnostics, or estimation details. While it is generally acknowledged that verifying assumptions like proportional hazards may be less critical for prediction-focused models, providing a clear description of how the model was developed, particularly regarding time-varying effects or parameter estimation methods, would greatly enhance its transparency and credibility.

The model’s mandatory use in the adult health check-up system of Taiwan emphasizes the importance of methodological openness. Currently, medical institutions must apply to access the official toolkits (e.g., API, component, or standalone versions) provided by the HPA [2]. In the absence of full-model specifications in the published paper, users are required to rely on these black-box tools, which limits independent risk calculation and verification. This dependency may inadvertently constrain clinical autonomy and academic scrutiny, especially when the model influences nationwide patient care and policy decisions.

In conclusion, while Chang et al. (2022) [1] made an important contribution to public health by developing a large-scale, data-driven prediction model, the current level of methodological detail leaves room for improvement. Clarifying the model type, explaining the derivation and interpretation of S_0_(10) for each endpoint, and providing information on statistical assumptions and estimation procedures would help align the study with international reporting standards such as TRIPOD. Given the model’s prominent role in the national health policy of Taiwan, greater transparency would not only support scientific reproducibility but also reinforce the confidence of clinicians, researchers, and the broader public in its application.

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
