# Peer review of "Methodological Considerations for a Risk Model Adopted into the Chronic Disease Prevention Policy of Taiwan. Comment on Chang et al. Developing and Validating Risk Scores for Predicting Major Cardiovascular Events Using Population Surveys Linked with Electronic Health Insurance Records. Int. J. Environ. Res. Public Health 2022, 19, 1319"

_ijerph, 2025, doi:10.3390/ijerph22071113_

Round 1
Reviewer 1 Report
Comments and Suggestions for Authors
This comment calls for transparent reporting of the prediction model decribed in the paper by HY Chang (hereafter abbreviated as HYC). I have fully read the paper by HYC and agree with the commentor that the reporting quality of the paper leaves much to be desired. In particular, the authors most likely have developed their prediction model with Cox regression, but this information cannot be found anywhere in the manuscript. There are only some minor points that I would like to add:
- HYC actually reported S_0(10) for MACE for both genders in the first paragraph of their results section. If these values of S_(10) are legit, the predicted 10-year risk for MACE can actually be calculated from the reported information in the paper. However, in the paper S_0(10) is interpreted as "The 10-year (MACE) event probabilities for men/women," which does not fit the definition of baseline survival function in Cox regressions, i.e. the survival function when all regressors are zero. Therefore, I would be very doubtful about if S_(10) is the genuine baseline survival function at year 10.
- Since the goal of the HYC paper is to develop a prediction model rather than to interpret the regression coefficients, whether proportional hazards is satisfied would not be of significant importance (the prediction performance would be more important). I would suggest that the commentor tone down on this line of argument.
- The commentor may consider citing the TRIPOD or TRIPOD + AI statement to reinfornce the requirement of transparent reporting in prediction models.
Author Response
|
Response to Reviewer 1 Comments
|
||
|
1. Summary |
|
|
|
We sincerely thank the reviewer for this thoughtful and constructive feedback. We appreciate the reviewer’s endorsement of the need for transparent reporting of the prediction model developed by Chang, H-Y et al. (2022), as well as the insightful suggestions to improve the clarity and focus of our comments. Below, we provide a point-by-point response to the reviewer’s comments, and all corresponding revisions are highlighted in the Track Changes function of WORD in the resubmitted manuscript. |
||
|
2. Questions for General Evaluation |
Reviewer’s Evaluation |
Response and Revisions |
|
Does the introduction provide sufficient background and include all relevant references? |
Yes |
(No changes needed.) |
|
Is the research design appropriate? |
Not applicable |
(No changes needed.) |
|
Are the methods adequately described? |
Not applicable |
(No changes needed.) |
|
Are the results clearly presented? |
Not applicable |
(No changes needed.) |
|
Are the conclusions supported by the results? |
Yes |
(No changes needed.) |
|
3. Point-by-point responses to Comments and Suggestions for Authors |
||
|
Comment 1: HYC actually reported Sâ‚€(10) for MACE for both genders in the first paragraph of the results section. If these values of S(10) are legit, the predicted 10-year risk for MACE can actually be calculated from the reported information in the paper. However, in the paper Sâ‚€(10) is interpreted as "The 10-year (MACE) event probabilities for men/women," which does not fit the definition of baseline survival function in Cox regressions, i.e., the survival function when all regressors are zero. Therefore, I would be very doubtful about if S(10) is the genuine baseline survival function at year 10.
|
||
|
Author response 1: Thank you very much for this insightful clarification. We fully agree with the reviewer’s observation. In our revised manuscript (Paragraph 3), we clarify that the values reported by HYC—although referred to as “10-year event probabilities”—appear to correspond more closely to population-level, event-free survival estimates rather than to the baseline survival function as defined in Cox regression models. We now state this distinction explicitly and explain why the definition used by HYC introduces ambiguity in interpreting or replicating absolute risk estimates.
Revised Text (Paragraph 3): |
||
|
Comments 2: Since the goal of the HYC paper is to develop a prediction model rather than to interpret the regression coefficients, whether proportional hazards is satisfied would not be of significant importance (the prediction performance would be more important). I would suggest that the commentor tone down on this line of argument. |
||
|
Author Response 2: We appreciate the reviewer’s important reminder about the distinction between explanatory and predictive modeling. In light of this, we have revised our comments to reflect a more appropriate tone. Specifically, in our discussion of statistical considerations (Paragraph 5), we have removed the emphasis on testing the proportional hazards assumption and instead focused on the need to describe the model-building process (e.g., time-varying effects, parameter estimation), which remains relevant even in prediction-focused models.
Revised Text (Paragraph 5): |
||
|
Comments 3: The commentor may consider citing the TRIPOD or TRIPOD + AI statement to reinforce the requirement of transparent reporting in prediction models. |
||
|
Author response 3: Thank you for this excellent suggestion. We have now cited the TRIPOD statement (Collins et al., 2015) in both the Introduction and the Conclusion to emphasize the importance of transparent reporting. |
||
|
4. Response to Comments on the Quality of English Language |
||
|
The reviewer did not raise any concerns about language quality. No changes were made. |
||
|
5. Additional clarifications |
||
|
We thank the reviewer again for the thoughtful and constructive input. We have implemented all suggestions with the aim of improving the clarity, tone, and relevance of our comments. We hope that the revised version provides a more balanced and constructive contribution to the academic discussion surrounding this influential model. |
||
